# Bootstrap-Conditioned Action Selection with Tabular Foundation Models

**Devansh Gupta** [1]   **Shiv Tavker** [1]   **Dmitry Efimov** [1]   **Suchitra Sathyanarayana** [1]   **Gitanjali Bhutani** [1]
**Boris N. Oreshkin** [1]

## Abstract

Contextual bandits offer a natural framework for sample-efficient personalization, but practical deployment remains difficult under sparse, biased interaction data, unreliable uncertainty estimates, and severe cold starts. We study whether pretrained tabular foundation models with in-context learning can be turned into randomized policies for online decision making. We propose BC-ICL (*Bootstrap-conditioned action selection using ICL*), which at each round draws a bootstrap resample of the interaction history, conditions a frozen pretrained ICL model on that resample, scores all actions, and selects the action with the highest sampled score. We further introduce an arm-context conditioning architecture that promotes shared statistical strength across actions and helps avoid common bootstrap failure modes of isolated-arm bandits. Empirically, this policy delivers strong early-round regret and regret performance on standard contextual bandit suites, outperforming established baselines under a strict online protocol.

## 1. Introduction

Personalized recommendation systems are naturally modeled as *contextual bandit* problems, in which each interaction consists of selecting an item based on observed user and item features and then observing an immediate reward such as a click or rating (Li et al., 2010). In practice, modern systems rely on a *single global policy* that maps contextual features to actions, rather than maintaining an independent bandit for each user. Per-user bandits are rarely viable at scale due to severe cold-start and data-efficiency issues; while global contextual policies achieve substantially higher sample efficiency and have become the standard abstraction in large-scale recommender systems and online

decision-making (Li et al., 2010; 2011; Chu et al., 2011; Lattimore & Szepesvári, 2020). In the global-policy setting, the main difficulty is to achieve principled exploration without sacrificing reward-model expressiveness. Classical contextual bandit methods typically assume linear reward models and rely on optimism or posterior sampling, which yields strong regret guarantees but limited representational power (Agrawal & Goyal, 2013; Russo et al., 2018). Kernelized bandits improve flexibility (Srinivas et al., 2010; Chowdhury & Gopalan, 2017), and neural contextual bandit methods can model substantially more complex reward functions (Riquelme et al., 2018; Zhou et al., 2020; Zhang et al., 2021). However, these neural methods must train task-specific networks from scratch using only bandit feedback. Their uncertainty estimates, often based on linearization or dropout, are also highly sensitive to hyperparameters and can be brittle precisely in the low-data and cold-start regimes where exploration matters most.

In parallel, recent advances in *in-context learning (ICL) foundation models for tabular data* have produced powerful predictors (Hollmann et al., 2025; Qu et al., 2025). These models adapt to new tasks from an in-context dataset, effectively amortizing Bayesian inference into a single forward pass. This progress raises a natural question: *can tabular ICL predictors be used directly for action selection in contextual bandits?* To address this question, we propose BC-ICL (*Bootstrap-conditioned action selection using ICL*). At each round, we draw a bootstrap resample of the observed interaction history, condition a frozen tabular ICL predictor on that resample, score all candidate actions, and play the action with the highest sampled score. This yields a randomized policy that reuses the pretrained inductive bias of the foundation model while introducing exploration through bootstrap-induced variation. We further introduce an arm-context conditioning architecture that promotes shared statistical strength across actions and helps avoid common bootstrap failure modes of isolated-arm bandit settings. Empirically, we instantiate BC-ICL with pretrained tabular ICL models such as TabPFN and TabICL and show strong regret performance across standard contextual bandit suites under a strict online protocol.

Our contributions are threefold: (1) We propose BC-ICL, a simple and practical contextual-bandit algorithm that cou-

[1]Amazon Science. Correspondence to: Devansh Gupta <devgpta@amazon.com>.

*Proceedings of the $2^{nd}$ ICML Workshop on Foundation Models for Structured Data*, Seoul, South Korea. 2026. Copyright 2026 by the author(s).

ples bootstrap resampling with a frozen in-context learning predictor, turning a pretrained supervised model into an exploration-capable decision rule. (2) We introduce a multiplicative arm-context feature map that enables shared exploration across actions through a common projected context representation. (3) We perform comprehensive empirical evaluation showing that BC-ICL, instantiated with TabPFN or TabICL, outperforms linear, kernelized, and neural contextual bandit baselines.

## 2. Method

We consider a stochastic contextual bandit with action set $\mathcal{A} = \{1, \ldots, K\}$. At round $t$, the learner observes a context $x_t \in \mathcal{X}$, selects an action $a_t \in \mathcal{A}$, and then observes a reward $r_t \in [0, 1]$ with conditional mean $\mu(x_t, a_t) = \mathbb{E}[r_t \mid x_t, a_t]$. The interaction history at round $t$ is $\mathcal{D}_t = \{(x_s, a_s, r_s)\}_{s=1}^{t-1}$. Our goal is to deploy a pretrained tabular ICL predictor as a frozen reward model inside this online decision problem. The key design choice is to induce exploration through randomized conditioning: rather than changing model parameters online, we randomize the conditioning history seen by the fixed backbone.

**Arm-context representation.** For each action $a$, let $e_a \in \mathbb{R}^{d_a}$ denote a fixed arm embedding, and let $P \in \mathbb{R}^{d_x \times d_a}$ be a shared projection matrix. We use the multiplicative arm-context feature map,

$$\Phi_{\text{mult}}(x, a) = \begin{bmatrix} x; \ e_a; \ (x^\top P) \odot e_a \end{bmatrix},$$

where $\odot$ denotes elementwise multiplication. This representation has three roles: (i) it preserves the raw context signal $x$, (ii) it encodes arm identity through $e_a$, and (iii) it introduces explicit arm–context interactions through the shared projected context $(x^\top P) \odot e_a$. The shared projection is important: perturbations of the conditioning history propagate through a common context representation and affect scores for multiple arms simultaneously, which is the key shared exploration mechanism.

**Bootstrap-conditioned action selection using ICL (BC-ICL).** Let $\mathcal{M}(\cdot \mid \mathcal{D})$ denote the pretrained ICL predictor conditioned on a dataset $\mathcal{D}$. At round $t$, we draw a bootstrap resample,

$$\widetilde{\mathcal{D}}_t \sim \text{Bootstrap}(\mathcal{D}_t),$$

obtained by sampling $|\mathcal{D}_t|$ observations with replacement from $\mathcal{D}_t$. We then define the sampled reward predictor,

$$\hat{\mu}_{\widetilde{\mathcal{D}}_t}(x, a) := \mathcal{M}\left( \Phi_{\text{mult}}(x, a) \mid \widetilde{\mathcal{D}}_t \right),$$

where model outputs are clipped to $[0, 1]$ when needed, so that predicted scores lie on the same scale as the bounded

rewards $r_t \in [0, 1]$. We evaluate all candidate actions under this predictor, and play

$$a_t \in \arg\max_{a \in \mathcal{A}} \hat{\mu}_{\widetilde{\mathcal{D}}_t}(x_t, a).$$

After observing $r_t$, we append $(x_t, a_t, r_t)$ to the history and continue. This policy induces a randomized decision rule through bootstrap-conditioned predictor draws.

## 3. Experiments

We evaluate BC-ICL along four axes: (i) How does BC-ICL compare to established linear, kernel, and neural bandit baselines? (ii) Is data-level bootstrapping necessary, or does the ICL model's internal uncertainty suffice for exploration? (iii) What is the computational cost of our approach? (iv) How does arm-context interaction affect regret?

**Datasets.** Following prior work on neural contextual bandits (Riquelme et al., 2018; Zhou et al., 2020), we convert supervised classification datasets into contextual bandit problems. At each round, the learner observes a feature vector $x_t$ and selects one of $K$ arms corresponding to class labels. The learner receives a binary reward $r_t = 1$ if the selected arm matches the true label, and $r_t = 0$ otherwise. We evaluate on eight datasets spanning a range of scales and difficulty levels: ADULT, COVERTYPE, ISOLET, LETTER, MUSHROOM, MAGIC TELESCOPE, and SHUTTLE from the UCI repository (Dua & Graff, 2017), as well as MNIST (LeCun et al., 2010).These datasets differ in dimensionality and number of arms, as well as in the degree to which rewards depend on interactions between context features and arm identity. For example, ISOLET provides a high-dimensional, multi-arm setting where modeling arm-context interactions is critical for fast learning. Dataset statistics are summarized in Table 3 of Appendix B.1. For MNIST and ISOLET, which exceed TabPFN's recommended feature limit of 500, we apply PCA preprocessing to retain 85% of the variance. We report results for the six most challenging benchmarks in the main text, and defer the remaining results to Appendix B.3.

**Baselines.** We compare against a comprehensive set of contextual bandit algorithms: **Random**: Uniform arm selection. **Linear TS** (Chu et al., 2011; Agrawal & Goyal, 2013): Linear reward models with Thompson Sampling. **LinUCB** (Li et al., 2010): Linear reward models with UCB. **Kernel UCB / TS** (Valko et al., 2013; Chowdhury & Gopalan, 2017): Kernelized reward models with RBF kernels. **NeuralUCB / NeuralTS** (Zhou et al., 2020; Zhang et al., 2021): Neural reward models with UCB and Thompson Sampling. **BootstrapNN** (Osband et al., 2016; Riquelme et al., 2018): Ensemble of neural networks trained on bootstrap samples.

| Algorithm | Covertype | Isolet | Letter | MNIST | Mushroom | Shuttle |
|---|---|---|---|---|---|---|
| Random | 8571 | 7497 | 9615 | 9000 | 4062 | 8571 |
| Linear TS | $4120.9 \pm 305.3$ | $4452.4 \pm 179.1$ | $7844.0 \pm 193.2$ | $1774.6 \pm 126.1$ | $720.7 \pm 18.5$ | $1611.3 \pm 30.7$ |
| LinUCB | $5752.4 \pm 200.3$ | $4799.1 \pm 180.8$ | $7717.2 \pm 150.9$ | $1568.7 \pm 100.5$ | $702.0 \pm 50.2$ | $1607.1 \pm 80.6$ |
| Kernel UCB | $3885.4 \pm 368.5$ | $5005.2 \pm 285.7$ | $7763.9 \pm 305.9$ | $2774.4 \pm 158.2$ | $313.0 \pm 66.6$ | $276.9 \pm 54.8$ |
| Kernel TS | $3583.9 \pm 117.2$ | $4966.7 \pm 166.9$ | $7290.6 \pm 265.8$ | $1810.0 \pm 171.1$ | $219.4 \pm 37.1$ | $172.3 \pm 37.1$ |
| NeuralUCB | $5127.5 \pm 52.3$ | $5345.0 \pm 344.4$ | $6780.4 \pm 199.9$ | $1656.3 \pm 42.2$ | $227.9 \pm 8.9$ | $354.4 \pm 11.4$ |
| NeuralTS | $5087.5 \pm 54.3$ | $4387.4 \pm 262.8$ | $6564.3 \pm 283.5$ | $1639.5 \pm 61.1$ | $252.3 \pm 62.5$ | $354.9 \pm 12.4$ |
| BootstrapNN | $3390.0 \pm 132.9$ | $5195.9 \pm 113.9$ | $5802.6 \pm 330.6$ | $1473.4 \pm 100.8$ | $130.1 \pm 14.5$ | $168.5 \pm 21.5$ |
| **BC-ICL-TabPFN** | $2751.6 \pm 27.8$ | $3575 \pm 332.1$ | $\mathbf{4074.2 \pm 355.1}$ | $\mathbf{1453.4 \pm 152.7}$ | $38.6 \pm 2.8$ | $106.4 \pm 7.7$ |
| **BC-ICL-TabICL** | $\mathbf{2630.9 \pm 56.3}$ | $\mathbf{2992.5 \pm 286.1}$ | $4108.4 \pm 258.4$ | $1936.8 \pm 187.3$ | $\mathbf{37.7 \pm 5.3}$ | $\mathbf{92.2 \pm 11.2}$ |

*Table 1.* Cumulative regret (mean $\pm$ standard deviation) over 10 seeds. Lower is better. Best results in **bold**. Adult and Magic Telescope results appear in Appendix B.3.

**Implementation Details.** We test BC-ICL with two ICL backbones: TabPFN or TabICL. We use the arm-context feature maps defined in Section 2 to construct inputs for the ICL model. By default, we use the multiplicative encoding $\Phi_{\text{mult}}(x, a) = [x; e_a; (x^\top P) \odot e_a]$, with $P$ set to a random matrix. This explicitly captures arm-context interactions via random projections. We set the embedding dimension $d_a = K$ (the number of arms) throughout our experiments. We ablate multiplicative encoding against the simpler one-hot baseline $\Phi_{\text{one-hot}}(x, a) = [x; e_a]$. For all baselines, we use the block-diagonal context representation standard in the neural bandit literature (Riquelme et al., 2018; Zhou et al., 2020): each arm $a$ is represented by a vector in $\mathbb{R}^{Kd}$ with the context $x$ placed in the $a$-th block and zeros elsewhere. This representation favors baselines by enabling fully arm-specific parameters. However, block-diagonal encoding is unsuitable for BC-ICL for two reasons: (1) the resulting dimensionality $Kd$ quickly exceeds TabPFN's input capacity (e.g., $26 \times 72 = 1{,}872$ features for ISOLET after PCA), and (2) the sparse, block-structured inputs diverge significantly from the dense tabular data distribution on which ICL models were pretrained. Our compact encodings remain well within TabPFN's limits while preserving dense feature structure. At each round, we draw a fresh bootstrap resample of the interaction history with replacement and condition the ICL model on it. Hyperparameters for all baselines follow Riquelme et al. (2018) and Zhou et al. (2020). Additional implementation details can be found in Appendix B.2.

**Main Results.** Table 1 presents cumulative regret across all datasets. BC-ICL achieves the lowest cumulative regret, with both BC-ICL-TabPFN and BC-ICL-TabICL substantially outperforming all baselines on most datasets. Our results demonstrate two complementary contributions. First, ICL backbones provide effective representations for contextual bandits: both TabPFN and TabICL versions substantially outperform non-ICL baselines on most datasets. Second, Bootstrap exploration improves naive ICL policies: Table 2 shows that bootstrap sampling yields 5–19% regret reduction over greedy and sampling baselines on harder

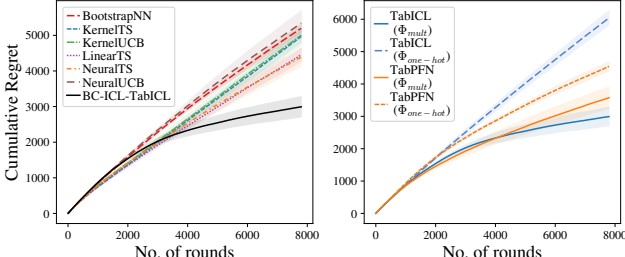

*Figure 1.* Cumulative regret on Isolet. **Left:** BC-ICL (TabICL) vs. baselines. **Right:** Ablation comparing multiplicative ($\Phi_{\text{mult}}$) vs. one-hot ($\Phi_{\text{one-hot}}$) arm-context interactions.

datasets (Covertype, ISOLET, Letter, MNIST), demonstrating that data-level bootstrapping enables effective exploration beyond the predictive uncertainty of ICL models alone. Key observations are as follows. **Strong gains over neural baselines.** On challenging multi-class datasets like ISOLET and Letter, our method substantially outperforms NeuralUCB, NeuralTS, and BootstrapNN, demonstrating that ICL backbones effectively leverage pretrained representations gaining an edge over networks trained from scratch online. Figure 1 (left) illustrates this on ISOLET, where BC-ICL-TabICL maintains consistently lower regret throughout learning. **Sample efficiency in small-data regimes.** On Mushroom, BC-ICL-TabICL achieves 85% lower regret than NeuralTS (37.7 vs. 252.3), highlighting the benefit of TabICL's pretrained inductive biases when online data is limited.

**Ablation Studies.** To confirm the effectiveness of the proposed BC-ICL we ablate it against two alternative strategies. **Greedy**: selects the arm with the highest predicted reward probability. **Sampling**: samples arms proportionally to predicted reward probabilities, i.e., selects arm $a$ with probability $\hat{p}(a \mid x_t)$ where $\hat{p}$ is the ICL model's softmax output. This leverages the model's predictive uncertainty for exploration. Table 2 confirms that predictive uncertainty alone is insufficient for sequential decision-making. On four of six datasets, greedy arm selection leads to substantially higher regret than BC-ICL, as the greedy policy commits

| Dataset | Greedy | Sampling | BC-ICL |
|---|---|---|---|
| Covertype | 2887 ± 330 | 2761 ± 285 | **2752 ± 28** |
| Isolet | 3777 ± 204 | 3589 ± 242 | **3575 ± 332** |
| Letter | 4481 ± 311 | 4342 ± 278 | **4074 ± 355** |
| MNIST | 1793 ± 133 | 1654 ± 146 | **1453 ± 153** |
| Mushroom | **36.1 ± 3.3** | 39.8 ± 4.1 | 38.6 ± 2.8 |
| Shuttle | **88.4 ± 17.2** | 94.2 ± 15.8 | 106.4 ± 7.7 |

*Table 2.* Ablation: cumulative regret (mean ± std) comparing exploration strategies with TabPFN. Smaller is better.

to suboptimal arms early and fails to recover. Sampling from the predictive distribution provides some improvement over greedy selection but still significantly underperforms compared to BC-ICL. This reveals that TabPFN's predictive uncertainties, while useful for i.i.d. prediction, may not capture the epistemic uncertainty required for exploration. On Mushroom and Shuttle, greedy selection performs comparably or better, consistent with the fact that exploration is less important on simpler problems. However, on more challenging multi-class datasets (ISOLET, Letter, MNIST), the gap between greedy/sampling and BC-ICL is substantial. **Multiplicative interactions accelerate learning.** Figure 1 (right) shows that the multiplicative feature map $\Phi_{\text{mult}}$ substantially outperforms one-hot encoding $\Phi_{\text{one-hot}}$ for both TabPFN and TabICL on ISOLET, reducing final regret by about 30–40%. This confirms that explicitly modeling arm-context interactions is critical, independent of the ICL backbone.

**Computational Cost.** A practical consideration for BC-ICL is computational efficiency. Unlike neural bandit methods that require gradient-based training at each round, ICL models perform prediction via a single forward pass, conditioning on the full interaction history. This becomes costly as the number of rounds grows. We therefore investigate two context selection strategies. **FIFO** retains the most recent window of 2,000 interactions. **KNN** retrieves the nearest neighbors of the current context $x_t$ from the full history (capped at 2,000 total). Implementation details are provided in Appendix B.4. Table 5 of Appendix B.4 reports average wall-clock time per round on ISOLET. Using the full context, BC-ICL-TabPFN is approximately 7× slower than neural baselines, while BC-ICL-TabICL is only 2× slower. With FIFO or KNN context selection, both methods become significantly faster: BC-ICL-TabICL (FIFO/KNN) achieves runtime competitive with BootstrapNN, while BC-ICL-TabPFN reduces from 4.02s to under 1.5s per round. Crucially, KNN-based context selection achieves regret comparable to using the full history (see Appendix B.4), making it an attractive choice for scaling BC-ICL to longer horizons without sacrificing performance.

## 4. Conclusion

We introduced BC-ICL, a method for turning pretrained tabular foundation models into randomized policies for contextual bandits via bootstrap-conditioned action selection. By resampling the interaction history and acting greedily with respect to the resulting predictor, BC-ICL induces exploration while preserving the inductive biases of in-context learning. Empirically, BC-ICL achieves strong performance across contextual bandit benchmarks, supporting the intuition that multiplicative arm-context representations enable useful statistical sharing across actions. BC-ICL assumes access to a pretrained tabular foundation model whose inductive biases are well aligned with the target task; performance may degrade under significant prior-task mismatch. From a practical standpoint, the bootstrap procedure introduces additional computational overhead due to repeated conditioning on resampled histories. Our experimental protocol uses fixed feature representations, and extensions that jointly adapt representation and uncertainty estimation online are not explored in this work.

On the theoretical side, under mild stability assumptions on the ICL predictor, one can derive a regret decomposition for BC-ICL. In particular, the expected regret could potentially be separated into a *stability term* that captures bootstrap concentration on the best arm when score gaps are large, and an additive *representation term* capturing prior-task mismatch together with on-path estimation error of the bootstrap-mean predictor. Formalizing this decomposition and characterizing the resulting regret rates remains an important direction for future work.

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

# A. Related Work

**Contextual Bandits.** Contextual bandit algorithms have been extensively studied as a framework for sequential decision-making with partial feedback (Lattimore & Szepesvári, 2020; Langford & Zhang, 2007). Linear contextual bandits assume a linear relationship between contexts and expected rewards, enabling algorithms with provable regret guarantees. LinUCB (Chu et al., 2011; Abbasi-Yadkori et al., 2011) uses optimism under uncertainty via confidence ellipsoids, while LinTS (Agrawal & Goyal, 2013) employs posterior sampling over linear reward parameters. These methods enjoy strong theoretical foundations but are limited in expressiveness and can perform poorly under model misspecification (Lattimore & Szepesvári, 2017). To address nonlinear reward structures, generalized linear bandits (Filippi et al., 2010) and kernelized bandits (GP-UCB (Srinivas et al., 2010) and KernelUCB (Valko et al., 2013; Chowdhury & Gopalan, 2017)) extend the linear framework using richer function classes and provide regret bounds under smoothness assumptions. Neural network-based contextual bandits further improve expressiveness by learning flexible reward models. NeuralUCB (Zhou et al., 2020) constructs confidence bounds using a neural tangent kernel approximation, while Neural Thompson Sampling by Zhang et al. (2021) extends posterior sampling to deep networks while Gal & Ghahramani (2016) consider dropout-based uncertainty. Most approaches rely on approximate uncertainty estimates that are sensitive to architectural choices and hyperparameters. Recent theoretical work highlights fundamental challenges in obtaining reliable exploration guarantees with overparameterized neural models in adaptive settings (Foster et al., 2021), particularly in low-data regimes. We address this challenge by starting from a pretrained tabular foundation model rather than a randomly initialized reward network, thereby importing a strong predictive prior into the bandit problem and improving robustness in data-constrained and cold-start environments.

**Bootstrap Methods for Exploration.** Bootstrap-based exploration offers a simple and scalable alternative to explicit Bayesian uncertainty quantification. Eckles & Kaptein (2014) propose bootstrap Thompson Sampling, which resamples observed data to approximate posterior uncertainty. Osband et al. (2016) introduce bootstrapped DQN for deep reinforcement learning, demonstrating that training an ensemble of networks on different bootstrap samples enables effective exploration without explicit posterior inference. Lu & Van Roy (2017) analyze ensemble sampling and establish connections to Thompson Sampling under certain conditions, while related work studies randomized exploration from a frequentist perspective (Kveton et al., 2019). Importantly, Kveton et al. (2019) analyze limitations of naive bootstrapping for exploration, showing that it can suffer linear regret in K-armed bandits without additional structure, motivating the shared-structure contextual bandit settings we focus on. Our work combines bootstrap-based exploration with pretrained ICL models: rather than training an ensemble from scratch, we apply bootstrap resampling to the conditioning set of a fixed pretrained backbone, inducing randomized exploration without online retraining.

**Generation-Based Exploration with Pretrained Models.** An alternative approach to exploration with pretrained models leverages generative capabilities to impute missing potential outcomes. Cai et al. (2024) propose active exploration via autoregressive generation of missing data, where a generative model trained on observational data is used to impute unobserved outcomes for all arms, enabling Thompson Sampling even for arms with no historical data. Zhang et al. (2025) extend this framework to contextual Thompson Sampling via generation of missing data, providing theoretical analysis and demonstrating strong empirical performance. These approaches directly address the cold-start problem by generating counterfactual outcomes for unobserved arms, enabling exploration in settings where no shared structure across arms can be assumed. BC-ICL is complementary to these generation-based methods. Our approach bootstraps only observed data, which is simpler and compatible with any ICL model that provides predictive capabilities, without requiring specialized generative modeling infrastructure.

**ICL Foundation Models for Tabular Data.** ICL has been studied as a form of implicit Bayesian or meta-learned inference in large transformer models (Xie et al., 2022; Garg et al., 2022). Prior-Data Fitted Networks (PFNs) (Hollmann et al., 2025) train transformers to approximate Bayesian inference by conditioning on datasets sampled from a structural causal model family. Other ICL approaches for tabular data include TabICL (Qu et al., 2025), which uses class-conditioned in-context learning, as well as retrieval-augmented methods that condition on relevant examples from large datastores. These models share a common paradigm: prediction is performed via a single forward pass conditioned on context data, making them computationally attractive for online and low-latency settings. In contrast, other tabular foundation models such as SAINT (Somepalli et al., 2021), TabTransformer (Huang et al., 2020) and NIAQUE (Oreshkin et al., 2025) require task-specific fine-tuning at test time. Meta-learning approaches for bandits and reinforcement learning amortize learning across tasks but still rely on gradient-based adaptation during deployment (Finn et al., 2017; Grant et al., 2018; Duan et al., 2016), making gradient-free inference ICL models particularly appealing for online settings.

# B. Experimental Details

## B.1. Dataset Details

Table 3 summarizes the characteristics of all datasets used in our experiments.

| Dataset | Rounds | Arms ($K$) | Features ($d$) |
|---|---|---|---|
| Adult (UCI) | 10,000 | 2 | 14 |
| Covertype (UCI) | 10,000 | 7 | 54 |
| Magic (UCI) | 10,000 | 2 | 10 |
| MNIST | 10,000 | 10 | 784 |
| Mushroom (UCI) | 8,124 | 2 | 22 |
| Shuttle (UCI) | 10,000 | 7 | 9 |
| Letter (UCI) | 10,000 | 26 | 16 |
| Isolet (UCI) | 7,797 | 26 | 617 |

*Table 3.* Dataset statistics.

For datasets where raw feature dimensionality exceeds TabPFN's recommended limit of 500 features (MNIST and ISOLET), we apply PCA to retain 85% of the variance, reducing dimensionality to 186 and 72 features, respectively.

## B.2. Implementation Details

For all experiments, we randomly shuffle all datasets and normalize each feature vector to have unit $\ell_2$ norm.

**Neural baselines.** All neural baselines (NeuralUCB, NeuralTS, BootstrapNN) use a single hidden layer with 100 neurons. Gradient descent is performed until convergence with learning rate $\eta = 0.001$. For BootstrapNN, we maintain an ensemble of $q = 10$ networks, with bootstrap probability $p = 0.8$ (i.e., each data point is included in each network's training set with probability 0.8) as is standard in the literature.

**Linear and kernel baselines.** For LinearTS, KernelTS, and KernelUCB methods, we set the regularization parameter $\lambda = 1$ and perform a grid search over the exploration parameter $\nu \in \{10^{-1}, 10^{-2}, 10^{-3}, 10^{-4}, 10^{-5}\}$. For NeuralTS and NeuralUCB, we perform a grid search over both $\lambda \in \{1, 10^{-1}, 10^{-2}, 10^{-3}\}$ and $\nu \in \{10^{-1}, 10^{-2}, 10^{-3}, 10^{-4}, 10^{-5}\}$.

**BC-ICL configuration.** We use the open-source TabPFNv2 and TabICL classifiers. For both TabPFN and TabICL, we set the number of estimators to $n_{\text{estimators}} = 8$, following the recommendation in the TabPFN documentation. While the default value for TabICL is $n_{\text{estimators}} = 32$, we found that reducing it to 8 significantly accelerates computation without degrading performance.

We also experimented with an alternative ensemble strategy: maintaining $n$ copies of the model, each with a single estimator ($n_{\text{estimators}} = 1$) and different preprocessing configurations, then randomly selecting one model for arm selection at each round. However, we observed that ICL models perform poorly with $n_{\text{estimators}} = 1$.

**PCA preprocessing.** For high-dimensional datasets exceeding TabPFN's recommended 500-feature limit (MNIST and ISOLET), we apply PCA to retain 85% of the variance, reducing dimensionality to 186 and 72 features, respectively. This follows the recommendation of the open-source TabPFNv2 implementation.

**Hardware.** All experiments were conducted on a single NVIDIA A10 GPU. While TabPFN natively supports multi-GPU parallelization, TabICL does not; for consistency, we use single-GPU execution for all methods.

## B.3. Additional Results

This section presents further empirical results complementing those reported in the main paper.

**Adult and Magic Telescope datasets.** Table 4 reports cumulative regret across the additional Adult and Magic Telescope datasets, and Table 6 reports the ablation results for different arm selection strategies.

| Algorithm | Adult | MagicTelescope |
|---|---|---|
| Random | 5000 | 5000 |
| Linear TS | $2230.3 \pm 25.9$ | $2684.9 \pm 39.2$ |
| Kernel UCB | $2140.9 \pm 115.9$ | $2464.2 \pm 41.3$ |
| Kernel TS | $2102 \pm 50.7$ | $2449.9 \pm 68.5$ |
| NeuralUCB | $2515.9 \pm 215.3$ | $2416.6 \pm 25.2$ |
| NeuralTS | $2413 \pm 33.8$ | $2390.6 \pm 47.7$ |
| BootstrapNN | $2162.9 \pm 76.1$ | $2415.9 \pm 74.4$ |
| **BC-ICL-TabPFN** | $1713.4 \pm 46.8$ | $1698.2 \pm 14.3$ |
| **BC-ICL-TabICL** | $\mathbf{1680.6 \pm 31.5}$ | $\mathbf{1526.5 \pm 31.1}$ |

*Table 4.* Cumulative regret (mean $\pm$ std over 10 seeds) for additional datasets. Lower is better.

| Algorithm | Time/Round (s) |
|---|---|
| Linear TS | 0.36 |
| NeuralTS | 0.57 |
| NeuralUCB | 0.57 |
| BootstrapNN | 0.61 |
| BC-ICL-TabPFN (full) | 4.02 |
| BC-ICL-TabPFN (FIFO) | 1.43 |
| BC-ICL-TabPFN (KNN) | 1.47 |
| BC-ICL-TabICL (full) | 1.10 |
| BC-ICL-TabICL (FIFO) | 0.63 |
| BC-ICL-TabICL (KNN) | 0.65 |

*Table 5.* Average wall-clock time per round (seconds) on Isolet.

| Dataset | Greedy | Sampling | BC-ICL |
|---|---|---|---|
| Adult | $1694 \pm 50.2$ | $1700 \pm 70.2$ | $\mathbf{1680.6 \pm 31.5}$ |
| MagicTelescope | $\mathbf{1519.5 \pm 31.8}$ | $1590.3 \pm 47.8$ | $1526.5 \pm 31.1$ |

*Table 6.* Ablation: cumulative regret comparing exploration strategies with TabPFN for additional datasets.

**Full regret curves.**    Figure 2 shows cumulative regret plots for all eight datasets. The left panel of each subfigure compares BC-ICL (BC-ICL-TabPFN or BC-ICL-TabICL) against baselines, while the right panel shows the ablation comparing multiplicative encoding ($\Phi_{\text{mult}}$) versus one-hot encoding ($\Phi_{\text{one-hot}}$) for arm-context interactions.

### B.4. Context Window and Computational Cost

In long-horizon bandit problems, conditioning the ICL model on the full interaction history becomes computationally expensive as the number of rounds grows. To address this, we investigate two context selection strategies that bound the conditioning set size while maintaining strong regret performance.

**FIFO Context Selection.**    In this strategy, we maintain a sliding window of recent interaction history. At each round, the current context is added to the conditioning set with bootstrap probability $p = 0.8$. When the number of stored interactions exceeds 2,000, the oldest contexts are discarded to ensure that the conditioning set contains at most the 2,000 most recent interactions. This approach bounds memory usage and ensures constant-time updates per round after 2,000 rounds.

**KNN-Based Context Selection.**    In this strategy, we retain the full interaction history but condition the model only on a subset selected at each round. Specifically, for the current context vector $x_t$, we retrieve nearest neighbors from the historical contexts using cosine similarity. The retrieved set is capped at a maximum of 2,000 interactions in total. The TabPFN or TabICL model is then conditioned only on this retrieved subset. This approach also ensures near constant-time updates per round after 2,000 rounds.

**Computational comparison.**    Table 5 summarizes the wall-clock time per round for all baselines and BC-ICL methods with different context selection strategies on the ISOLET dataset. Using the full context, BC-ICL-TabPFN is approximately $7\times$ slower than neural baselines, while BC-ICL-TabICL is only $2\times$ slower. With FIFO or KNN context selection, both methods become significantly faster: BC-ICL-TabICL (FIFO/KNN) achieves runtime competitive with BootstrapNN, while BC-ICL-TabPFN reduces from 4.02s to under 1.5s per round.

**Regret performance with context selection.**    Figure 3 shows cumulative regret plots for all datasets with different context selection strategies. Crucially, KNN-based context selection achieves regret comparable to using the full history, making it an attractive choice for scaling BC-ICL to longer horizons without sacrificing performance. FIFO also performs well in practice, though it can suffer minor degradation on some datasets where older examples remain informative.

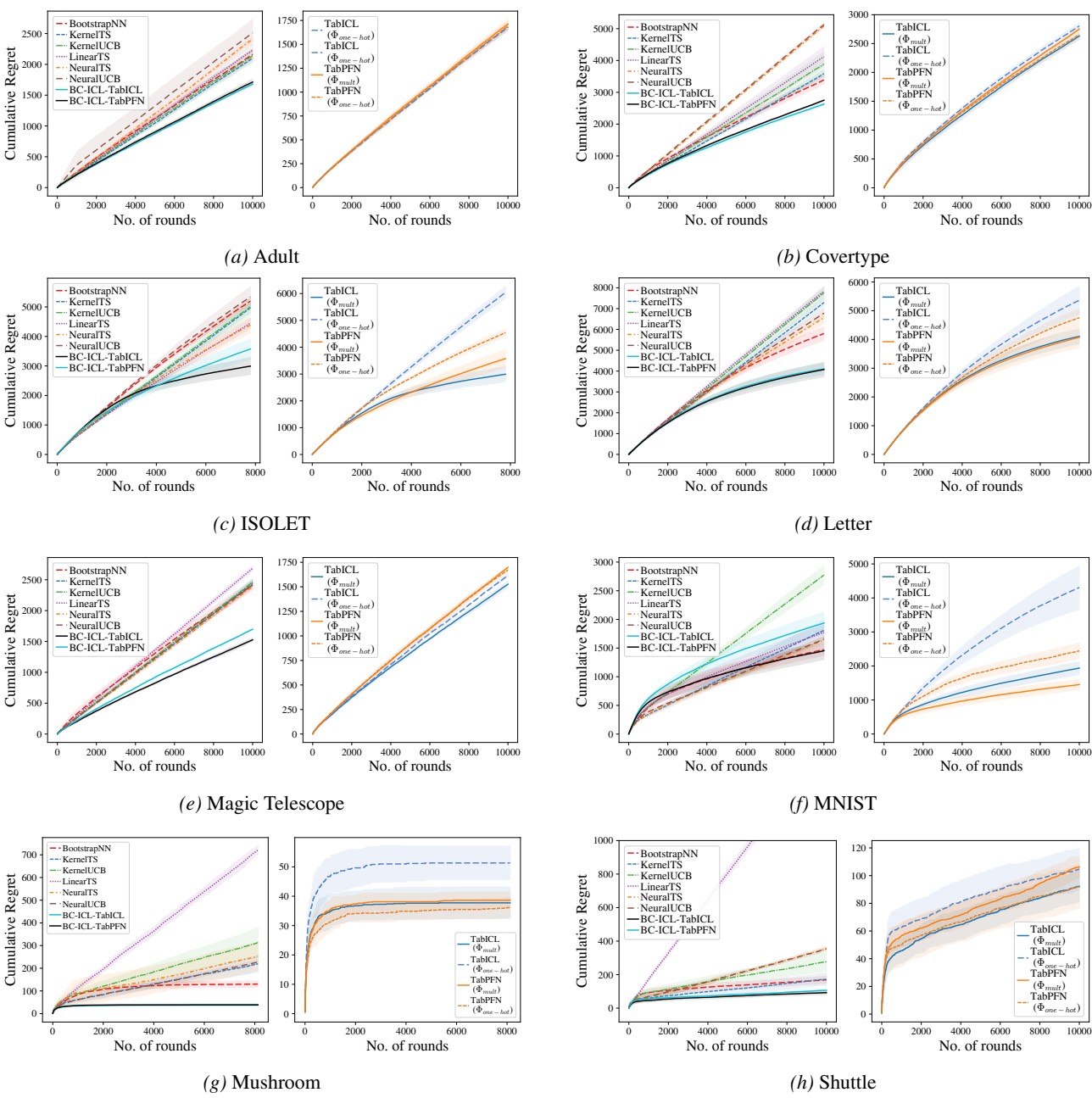

*Figure 2.* Cumulative regret across all datasets. **Left:** BC-ICL vs. baselines. **Right:** Ablation comparing multiplicative ($\Phi_{\text{mult}}$) vs. one-hot ($\Phi_{\text{one-hot}}$) arm-context interactions.

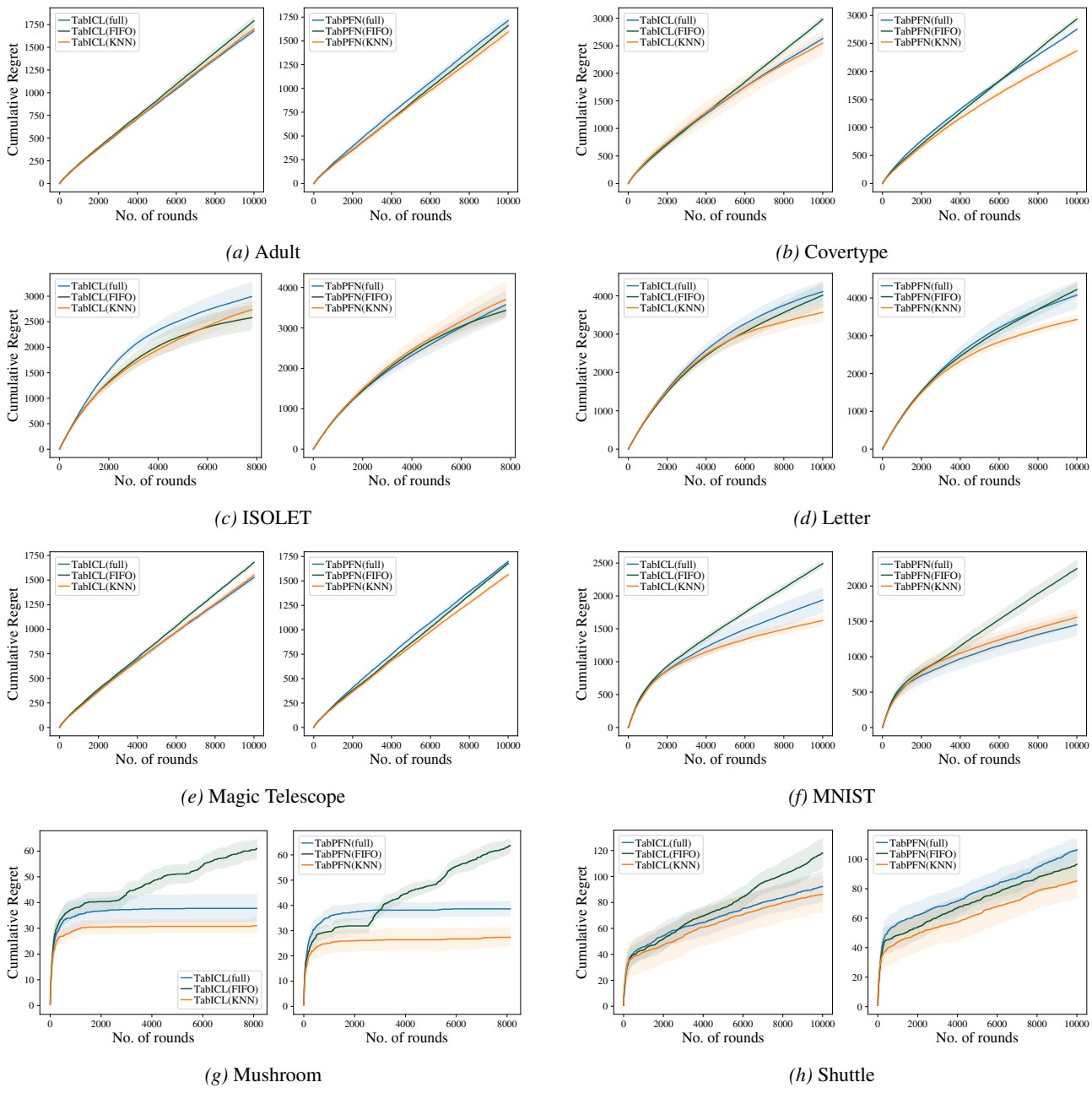

*Figure 3.* Cumulative regret with different context selection strategies. **Left:** TabICL. **Right:** TabPFN.

