# OpenReview forum: "Bootstrap-Conditioned Action Selection with Tabular Foundation Models"
_ICML.cc/2026/Workshop/FMSD — FMSD @ ICML 2026 Poster_

### Official Review · Reviewer_p3Nm · 2026-05-21
**Fair contribution but further analysis required**

**Rating:** 6
**Confidence:** 3

**Review:**

This paper applies tabular foundation models (TabPFN and TabICL) to sequential decision-making tasks. In particular, it focuses on the contextual bandit setting, where pretrained models are used to replace commonly used reward predictors (linear or neural network-based models). In a nutshell, at each step and for each arm, a subset of histories is subsampled with bootstrapping. Each subset draw is fed to the predictor to obtain the reward distribution, which is then used to select the arm for this step. In addition to the raw information (arm and context), the paper uses a feature embedding based on a linear projection to boost performance. As experimental validation, the authors show that the proposed method improves over existing contextual bandit baselines. An ablation study only motivates the merits of feature embedding.

The proposed idea is simple yet provides strong empirical performance. The idea of applying existing tabular foundation models to bandit problems is itself interesting and new to me. The paper is well written, clear, and to the point.

Despite the strengths, I have a few comments that the authors might want to incorporate for the final version.

- It would be good to isolate the assessment of the quality of predictors (both mean and uncertainty) from the regret curves. This might provide better insights on what leads to performance improvement.
- It remains unclear why the random matrix feature mapping is efficient. The authors state that this mapping helps to "avoid common bootstrap failure modes of isolated-arm", but do not provide clear evidence how. Additionally, it is unclear if $P$ is sampled at each new round t or per run (fixed for all rounds).
- Could you elaborate how BC-ICL-* addresses the cold start problem? According to section 2, the proposed method picks the arm with highest predicted mean reward at each round, which thus discards exploration as uncertainties are not considered. In my understanding, the exploration is mostly driven by the bootstrapping. However, without enough initial random exploration, the bootstrapping resample could collapse.

Overall, I find the contribution is sound, but the paper still benefits from better analysis to fully consolidate the results and findings.